

# Fructose 1,6-bisphosphatase 1 is a potential biomarker affecting the malignant phenotype and aerobic glycolysis in glioblastoma

Weihong Lu[1], Guozheng Huang[2], Yihan Yu[1], Xia Zhai[1] and Xiangfeng Zhou[3]

[1] Medical Molecular Biology Laboratory, School of Medicine, Jinhua University of Vocational Technology, Jinhua, China
[2] Department of Quality Management, Jinhua Fifth Hospital, Jinhua, China
[3] Clinical Medicine Department, School of Medicine, Jinhua University of Vocational Technology, Jinhua, China

Corresponding authors
Xia Zhai, 20211049@jhc.edu.cn
Xiangfeng Zhou, zxfjhc@163.com

## ABSTRACT

**Background:** Fructose 1,6-bisphosphatase 1 (*FBP1*) has been considered as a potential prognostic biomarker in glioblastoma (GBM), and this study explored the underlying mechanism.
**Methods:** The expression and effect of *FBP1* expression on the prognosis of GBM patients were examined applying bioinformatics analyses. After measuring the expression of *FBP1* in normal glial cell line HEB and GBM cells, cell counting kit-8 (CCK-8), 5-ethynyl-2-deoxyuridine (EdU), colony formation, transwell, and wound healing assay were carried out to examine the effects of silencing *FBP1* on the proliferation and invasion of GBM cells. Aerobic glycolysis was measured by calculating the extracellular acidification rate (ECAR) and oxygen consumption rate (OCR) of *FBP1*-silenced GBM cells. Furthermore, the protein levels of the mediators related to PI3K/AKT pathway and *BCL2* protein family were detected *via* immunoblotting. Additionally, the effects of *FBP1* silencing on the macrophage M2 polarization were assessed based on the fluorescence intensity of *CD206* and the phosphorylation of *STAT6* quantified by immunofluorescence and immunoblotting, respectively.
**Results:** High-expressed *FBP1* was indicative of a worse prognosis of GBM. *FBP1* knockdown in GBM cells suppressed the proliferation, invasion, migration, and aerobic glycolysis of GBM cells, lowered the phosphorylation levels of *AKT* and *PI3K* and the protein expression of *BCL2* but promoted *BAX* protein expression. Moreover, *FBP1* knockdown reduced CD206 fluorescence intensity and the phosphorylation of STAT6.
**Conclusion:** To conclude, *FBP1* could be considered as a biomarker that affected the malignant phenotypes and aerobic glycolysis in GBM, contributing to the diagnosis and treatment of GBM.

# INTRODUCTION

Glioblastoma (GBM) is a central nervous system cancer that is characterized by dynamic adaptation and subclonal diversity and accounts for around 49% of malignant brain tumors (*Ravi et al., 2022*; *Schaff & Mellinghoff, 2023*; *Zhang et al., 2024*). Currently, the standard treatment for GBM has remained unchanged since the introduction of adjuvant temozolomide, and the prognosis of GBM patients is largely unfavorable (*Ma, Taphoorn & Plaha, 2021*; *Hieu et al., 2024*). Although GBM has been widely studied in the fields of genetics and cell biology but the treatments for the cancer have not been revolutionized (*Ramanathan & Lorimer, 2022*; *Precilla et al., 2022*). Hence, to elucidate the molecular mechanism of the heterogeneity of GBM and to identify accurate targets are of great clinical significance to improve GBM therapies.

Existing studies have extensively discussed the roles and effects of genes knockdown or knockout on the malignant phenotypes of GBM cells (*Kurdi et al., 2023*). For instance, Brahma-related gene-1 (*BRG1*) is a mutually exclusive catalytic subunit that opens or closes chromatin to modulate gene transcription and promotes the malignant phenotype of GBM cells (*Wang et al., 2021*). Nuclear receptor-binding protein 1 (*NRBP1*) could enhance the malignant phenotypes of GBM through activating the PI3K/AKT pathway (*Zhang et al., 2024*). At the same time, biomass and adenosine triphosphate (ATP) in GBM are synthesized by fermentation metabolism, independent of intratumoral cellular or genetic heterogeneity (*Seyfried et al., 2022*). *PRMT3* is a member of protein arginine methyltransferases (PRMTs) with a high expression in GBM, and *PRMT3* knockdown drives the progression of GBM through enhancing glycolytic metabolism (*Liao et al., 2022*). Also, the POU class homeobox 2 (*POU2F2*) can modulate the glycolytic reprogramming and GBM development through phosphoinositide-dependent kinase-1 (*PDPK1*)-dependent PI3K/AKT/mechanistic target of rapamycin (mTOR) pathway (*Yang et al., 2021*). These findings encouraged us to discover the targets that modulate the malignant phenotypes and aerobic glycolysis of GBM cells.

Fructose 1,6-bisphosphatase 1 (*FBP1*) is an important enzyme implicated in the gluconeogenesis pathway and is expressed in most cells. *FBP1* functions critically in regulating gluconeogenesis and the synthesis of glycogen and other polysaccharide substances (*Wang et al., 2023*). Recent studies have stressed the role of *FBP1* as a novel proto-oncogene with pivotal functions in tumorigenesis (*Li et al., 2024*). Notably, *FBP1* knockdown inhibits the formation of ovarian cancer and cisplatin resistance *via EZH2*-mediated H3K27me3 (*Xiong et al., 2022*). Meanwhile, *FBP1* knockdown promotes the radiosensitivity of prostate cancer cells *via* promoting autophagy (*Li et al., 2020*). Furthermore, *FBP1* could dephosphorylate IκBα and suppress colorectal tumorigenesis (*Zhu et al., 2023*). In GBM, study reported that downregulating *FBP1* expression could rewire the metabolic processes and affect the aggressiveness of GBM (*Son et al., 2020*). These results provided a basis for the current study to explore the underlying mechanisms of the role of *FBP1* in GBM.

This work examined the potential role and mechanism of *FBP1* in GBM. *FBP1* was silenced using the small interfering RNA (siRNA) technique, and the effects on GBM cell

proliferation, invasion, glucose metabolism, as well as the PI3K/AKT pathway and the BCL2/BAX protein family were comprehensively assessed. Additionally, we also explored the regulatory effects of *FBP1* silencing on macrophage M2-type polarization. The study aimed to elucidate the biological functions of *FBP1* in GBM and to reveal its potential role in the tumor metabolism and immune microenvironment. We propose that *FBP1* can serve as a novel target for GBM diagnosis and treatment. These findings contribute to advancing precision treatment strategies for GBM.

## MATERIALS AND METHODS

### Data acquisition and sources for bioinformatics analysis

*FBP1* transcription levels in a variety of cancers were determined based on the *FBP1* expression data obtained from The Cancer Genome Atlas (TCGA) and Genotype-Tissue Expression (GTEx). *FBP1* expression levels in GBM and normal tissues were compared in the GEPIA2 (http://gepia2.cancer-pku.cn/#index) platform and visualized into histograms. Disease recurrence-free survival analysis was conducted by Kaplan-Meier survival curves to assess the survival differences between high and low *FBP1* expression groups, and risk ratios (HR) were calculated using log-rank test and Cox risk regression model.

### Cell culture and intervention

The 90% high-glucose Dulbecco's modified Eagle's medium (DMEM, C0891; Beyotime, Shanghai, China) with 10% fetal bovine serum (FBS, C0234; Beyotime, Shanghai, China) was used to culture human GBM cell lines U87 (BNCC340411) and U251 (BNCC100123) obtained from BeiNa culture Bio (Xinyang, China). Meanwhile, human monocyte cell line THP-1 (IM-H260) and normal glial cell line HEB (IM-H209) were obtained from ImmoCell (Xiamen, China). Roswell Park Memorial Institute-1640 medium (C0893; Beyotime, Shanghai, China) containing 10% FBS, 1% penicillin-streptomycin, and 0.05 mM 2-mercaptoethanol (M27072; Acmec Biochemical, Shanghai, China) was used to culture the THP-1 cells, whereas the HEB cells were grown in normal DMEM (D6540; Solarbio, Beijing, China) added with 1% penicillin-streptomycin and 10% FBS (C0222; Beyotime, Shanghai, China). All the cells were incubated in the incubator in 5% $CO_2$ with at 37 °C. In this study, the cell lines have been authenticated by STR profiling, and mycoplasma contamination was excluded as the mycoplasma testing results were negative.

For subsequent studies, monocyte cell line THP-1 was treated with 50 ng/mL phorbol-12-myristate-13-acetate (PMA, P6741; Solarbio, Beijing, China) and the induced macrophages were co-cultured with GBM cell line for subsequent assays (*Gatto et al., 2017*).

Meanwhile, the siRNAs targeting *FBP1* (si-*FBP1#1*, si-*FBP1#2*) and corresponding negative control siRNA (si-NC) were purchased from GenePharma (Shanghai, China) and transfected into GBM cells utilizing lipofectamine 2000 transfection reagent (11668-027; Invitrogen, Carlsbad, CA, USA), following the manuals. The sequences for transfection were listed in Table 1.

**Table 1 Sequences for the knockdown assay.**

| Target | Sequence (5′-3′) |
| --- | --- |
| si-NC | ACCTTGTGAACAGGTTAGTTAAA |
| si-FBP1#1 | ACCTGGTTATGAACATGTTAAAG |
| si-FBP1#2 | TGGTTATGAACATGTTAAAGTCA |

## Cell proliferation assays

Cell counting kit-8 (CCK-8) and colony formation assay were carried out to measure the GBM cell proliferation. In detail, the transfected GBM cells at a density of $2 \times 10^3$ cells/well were planted into the 96-well plates and cultured for 12, 24 and 36 h (h). Next, 10 μL CCK-8 reagent (C0037; Beyotime, Shanghai, China) was added for 1-h incubation. A microplate reader (iMark; Bio-Rad, Hercules, CA, USA) was employed to read the OD at 450 nm (*Feng & Xiao, 2024*).

The transfected U87 and U251 cells were cultured for 48 h to the logarithmic growth phase and then transferred into 96-well plates. The cell proliferation was measured by the EdU Cell Proliferation Assay Kit (RiboBio, Guangzhou, China). Following the protocol, the cells were stained, examined and photographed under a fluorescence microscope (Nikon, Tokyo, Japan). The counting of EdU-positive cells was performed with ImageJ software.

To perform the colony formation assay, the transfected GBM cells were inoculated into the 6-well plate (400 cells/well) and maintained for 14 days. Next, the colonies formed were fixed by 4% paraformaldehyde (P0099; Beyotime, Shanghai, China) for 30 minutes (min) and dyed with 0.1% crystal violet (C0121; Beyotime, Shanghai, China). Finally, the colonies were quantified for additional analyses.

## Cell invasion assay

GBM cells ($5 \times 10^5$ cells) were plated on the upper polycarbonate Transwell filter precoated with Matrigel (C0371; Beyotime, Shanghai, China) of a 24-well cell invasion assay chamber (3422; Corning, Inc., Corning, NY, USA), which was pre-filled with 200 μL serum-free medium. Meanwhile, 700 μL medium with 10% serum was supplemented into the lower chamber as the chemoattractant. After incubating the GBM cells at 37 °C for 48 h, non-invasive cells were removed carefully using cotton swabs, while invaded cells were further fixed in 4% paraformaldehyde for 10 min, dyed with crystal violet and counted under the optical microscope (DP27; Olympus, Tokyo, Japan).

## Wound healing assays

Wound healing assay was conducted to measure the cell migration capacity. In brief, the cells ($5 \times 10^5$ cells/well) were allowed to reach full confluency in 6-well plates. After the creation of a scratch wound with a 10 mL pipette tip, serum-free medium was used for cell incubation for 24 h at 37 °C in 5% $CO_2$. The migrating cells were observed and photographed using a light microscope.

## Flow cytometry

U87 and U251 cells transfected with FBP1-specific siRNA (si-FBP1) or negative control (si-NC) were collected, rinsed in PBS, and then resuspended in 195 μL of annexin-V FITC (BD Biosciences, Franklin Lakes, NJ, USA) containing 5 μL of propidium iodide (PI), according to the instructions. Subsequently, after 10-min cell incubation in the dark at room temperature, flow cytometry was utilized for the analysis. The data were evaluated in Lysis software (EPICS-XL, Ramsey, MN, USA).

## Cell glycolysis assay

The ECAR and OCR were calculated in the Seahorse XFe24 Analyzer (Agilent, Santa Clara, CA, USA) to reflect the aerobic glycolysis of GBM cells (*Nguyen et al., 2021*). Specifically, the transfected GBM cells were inoculated into the XFe24 cell culture microplates of the analyzer at a concentration of $3 \times 10^4$ cells/well or into the XFp cell culture microplates at a density of $5 \times 10^3$ cells/well in the culture medium added with 10% FBS, 5 mM D-glucose (ST1227; Beyotime, Shanghai, China) and 1 mM pyruvic acid (P15930; Acmec Biochemical, Shanghai, China) overnight. The GBM cells were treated with reagents containing the medium of 5 mM D-glucose, 1 mM pyruvic acid and 1.5% FBS the next day. Seahorse XF base medium (102353-100; Agilent, Santa Clara, CA, USA) added with 10 mM D-glucose, 2 mM glutamine (L51430; Acmec Biochemical, Shanghai, China) and 1 mM pyruvic acid was prepared for mitochondrial stress assay (103015-100; Agilent, Santa Clara, CA, USA). For mitochondrial stress assay, 2 μM oligomycin (O24350; Acmec Biochemical, Shanghai, China), 2 μM carbonyl cyanide-4 (trifluoromethoxy) phenylhydrazone (FCCP, F17810; Acmec Biochemical, Shanghai, China), 0.5 μM rotenone (R70211; Acmec Biochemical, Shanghai, China) and 0.5 μM antimycin A (A8674; Sigma, St. Louis, MO, USA) were added in sequence. The Seahorse XF base medium with 1 mM L-glutamine was applied for glycolysis stress assay and sequentially supplemented with 10 mM D-glucose, 1 μM oligomycin and 50 mM 2-Deoxy-D-glucose (ST1024; Beyotime, Shanghai, China). The corresponding results were calculated and plotted within 2 h.

## Immunofluorescence

Following the co-culture, the induced macrophages were fixed by 4% paraformaldehyde for 15 min and permeabilized by 0.25% Triton X-100 (P0096; Beyotime, Shanghai, China) at ambient temperature. Next, the macrophages were blocked using 1% bovine serum albumin (ST023; Beyotime, Shanghai, China) and further incubated with the following antibodies: Allophycocyanin-labeled anti-F4/80 antibody (MF48005, 1:500; Invitrogen, Carlsbad, CA, USA), Alexa Fluor® 488-labeled anti-CD206 antibody (ab313398, 1:500; Abcam, Cambridge, UK), goat anti-rabbit IgG (ab6702, 1:500; Abcam, Cambridge, UK) and goat anti-mouse IgG (ab6708, 1:500; Abcam, Cambridge, UK). After rinsing, the cell nuclei were dyed with DAPI (C1002; Beyotime, Shanghai, China). The corresponding images were taken using a fluorescence microscope (ECLIPSE E800; Nikon, Tokyo, Japan).

## RNA extraction and qRT-PCR

TRIzol reagent (15596-026; Invitrogen, Carlsbad, CA, USA) was applied to separate the total RNA according to the protocol and the complementary DNA was synthesized with a commercial synthesis kit (D7170S; Beyotime, Shanghai, China). QRT-PCR was then conducted together with the SYBR Green qPCR Mix (D7260; Beyotime, Shanghai, China) and ABI7500 thermocycler (ThermoFisher Scientific, Waltham, MA, USA). The conditions for the PCR reaction were set as follows: 30 seconds (s) at 95 °C, followed by 10 s at 95 °C, then 30 s at 60 °C, and finally 34 s at 70 °C, for a total of 40 cycles. The relative mRNA levels were calculated by the method $2^{-\Delta\Delta CT}$, with β-actin as a housekeeping control (*Livak & Schmittgen, 2001*). See Table 2 for the sequences of primers (*Sindhuja, Amuthalakshmi & Nalini, 2022*).

## Immunoblotting

Whole cells were lysed using the RIPA lysis buffer (P0013B; Beyotime, Shanghai, China) and the concentration of lysed protein sample was tested. Hereafter, the samples were separated on the separation gel and moved onto the polyvinylidene fluoride membranes (*Zhang et al., 2023*), which were then blocked using 5% non-fat milk and incubated together with the antibodies against phospho-PI3K (#4228, 1:1000; Cell Signaling Technology, Danvers, MA, USA), PI3K (#4292, 1:1000; Cell Signaling Technology, Danvers, MA, USA), phospho-AKT (ab38449, 1:1000; Abcam, Cambridge, UK), STAT6 (#9362, 1:1000; Cell Signaling Technology, Danvers, MA, USA), BCL-2 associated X (BAX, ab32503, 1:2000; Abcam, Cambridge, UK), BCL2 (ab182858, 1:2000; Abcam, Cambridge, UK), AKT (ab8805, 1:2000; Abcam, Cambridge, UK), phospho-signal transducers and activators of transcription 6 (STAT6, #9361, 1:1000; Cell Signaling Technology, Danvers, MA, USA), and housekeeping control β-actin (ab8226, 1:1000; Abcam, Cambridge, UK) overnight at 4 °C. Then the horseradish peroxide-conjugated goat anti-rabbit IgG (ab205718, 1:5000; Abcam, Cambridge, UK) and anti-mouse IgG (ab205719, 1:5000; Abcam, Cambridge, UK) secondary antibodies were reacted with the membrane for 1 h at ambient temperature. Then TBST was used to wash the membranes, which were then treated with the visualization reagent (P0018S; Beyotime, Shanghai, China) and visualized by ChemiDoc imaging system (Bio-Rad, Hercules, CA, USA). The densitometry analysis was conducted in ImageJ software (v. 1.42; National Institutes of Health, Bethesda, MD, USA).

## Statistical analyses

All experiments were independently performed for three times and the statistics were analyzed in GraphPad Prism software (v. 8.0.2; GraphPad, Inc., La Jolla, CA, USA). Continuous data involving two groups were analyzed by unpaired t-test and shown as mean ± standard deviation. A *p*-value < 0.05 signified that the data were statistically significant.

**Table 2 Sequences of Primers for PCR.**

| Target | Sequence (5′-3′) |
| --- | --- |
| FBP1 | |
| Forward | GGGTAAATATGTGGTCTGTT |
| Reverse | AACTTCTTCCTCTGGATGTA |
| β-actin | |
| Forward | CACACCTTCTACAATGAGC |
| Reverse | ATAGCACAGCCTGGATAG |

# RESULTS

## Bioinformatics analysis of *FBP1* in GBM

Analysis on the role of *FBP1* in GBM based on the relevant data and the pan-cancer level of *FBP1* revealed a relatively higher level of *FBP1*, especially in GBM (Figs. 1A and 1B, $p < 0.05$). At the same time, high *FBP1* level was related to a poor cancer prognosis (Fig. 1C). These results suggested a higher level of *FBP1* in GBM and the association between high-expressed *FBP1* and worse prognosis of GBM patients.

## Quantification of *FBP1* level in GBM

In GBM cells and normal glial cell line HEB, the data of qRT-PCR (Figs. 2A and 2B) and immunoblotting (Figs. 2C and 2D) demonstrated that *FBP1* had remarkably higher expression in GBM cells than in HEB cell line (Figs. 2A–2D, $p < 0.05$). Then siRNA against *FBP1* was synthesized and transfected into GBM cells to test the knockdown efficiency. In accordance with the results, the 2nd siRNA with a better knockdown efficiency was applied for subsequent studies (Figs. 2E and 2F, $p < 0.05$). Here, an abnormally high level of *FBP1* in GBM was observed, providing evidence for subsequent *in vitro* assays.

## Effect of FBP1 silencing on the proliferation, migration, invasion, apoptosis, and aerobic glycolysis of GBM cells

As shown in Figs. 3A and 3B, we observed that silencing of *FBP1* greatly suppressed the proliferation of U251 and U87 cells. Additionally, the number of clones was noticeably more in the control group than in the si-*FBP1* group (Fig. 3C, $p < 0.05$). Also, transwell and wound healing assays also showed that silencing of *FBP1* remarkably lowered the invasive and migratory capacity of GBM cells (Figs. 4A and 4B, $p < 0.05$). The si-*FBP1* group had a notably elevated apoptosis level of GBM cells than the control group (Fig. 4C).

The ECAR and OCR of GBM cells after the silencing of *FBP1* were additionally calculated to reflect the aerobic glycolysis. It was found that *FBP1* silencing lowered ECAR but increased OCR in GBM cells U87 and U251 (Figs. 5A–5D). Collectively, these results indicated that knocking down *FBP1* suppressed the malignant phenotype and aerobic glycolysis of GBM cells.

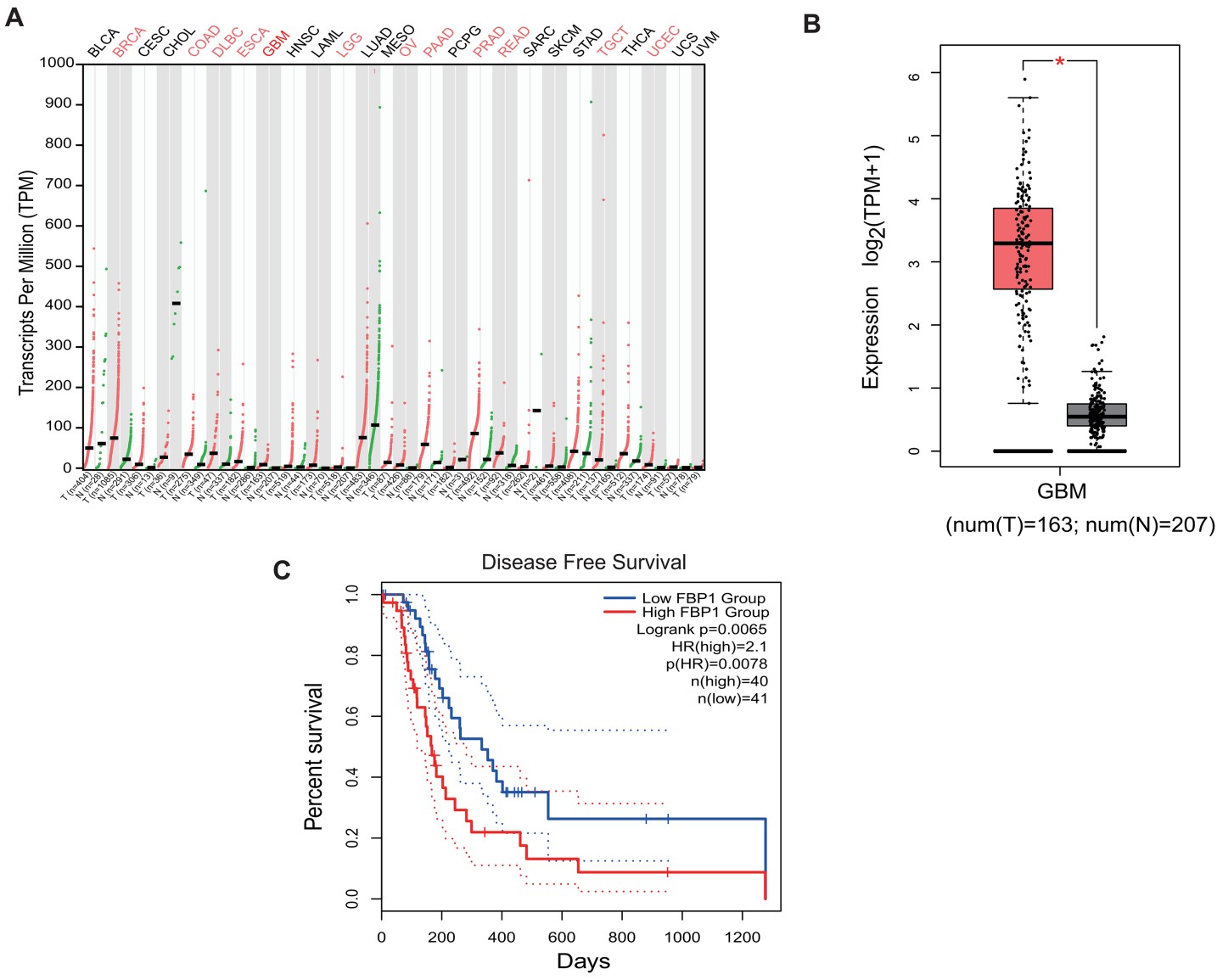

**Figure 1 Bioinformatics on *FBP1* in glioblastoma.** (A) Pan-cancer *FBP1* expression data from TCGA. (B) *FBP1* expression level in glioblastoma, based on the data from GEPIA 2. (C) Impact of high/low *FBP1* expression level in survival of glioblastoma patients. **p* < 0.05.

## Effects of *FBP1* silencing on the mediators related to PI3K/AKT pathway in GBM cells

The effect of *FBP1* silencing on the protein expression of the pathway was assessed by Western blot to further analyze whether *FBP1* affected GBM cells by regulating the PI3K/AKT pathway. The result demonstrated that the phosphorylation levels of *AKT* and *PI3K* reduced significantly after *FBP1* silencing in U251 and U87 cells, respectively ($p < 0.05$) (Figs. 6A–6D). These findings indicated that *FBP1* played a crucial role in GBM cells through regulating the PI3K/AKT pathway.

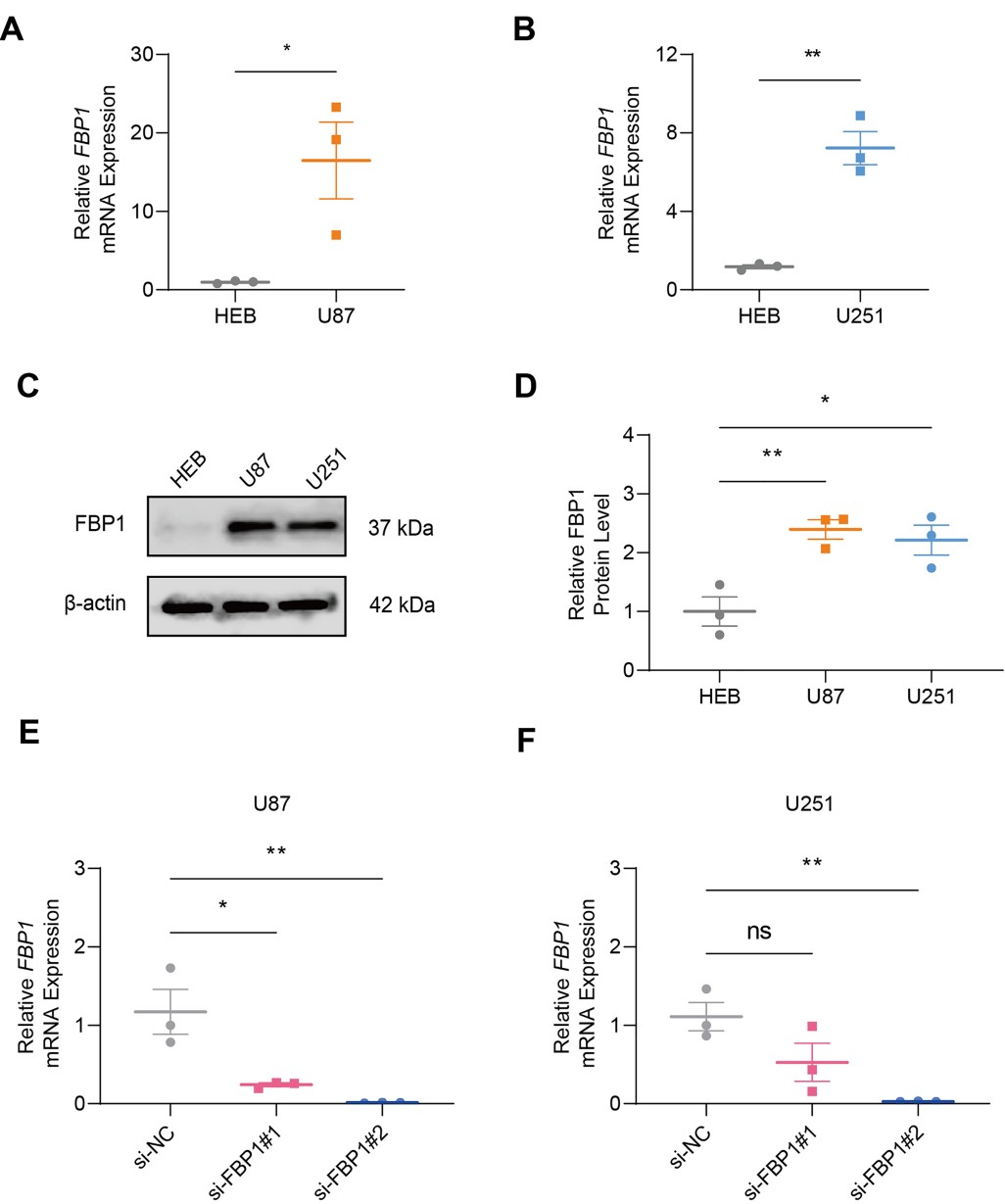

**Figure 2 Quantification of *FBP1* level in glioblastoma.** (A and B) Calculated *FBP1* mRNA expression in normal glial cell line HEB and glioblastoma cell line U87 (A) and U251 (B). (C and D) Calculated *FBP1* protein expression in normal glial cell line HEB and glioblastoma cell line U87 and U251. (E and F) Validation on the knockdown efficiency of *FBP1*-specific small interfering RNA in glioblastoma cell line U87 (E) and U251 (F). All data of three independent trials were expressed as mean ± standard deviation. The statistical significance was shown as either the asterisks (*$p < 0.05$, **$p < 0.01$) or the caption "ns" ($p > 0.05$).

## Effect of FBP1 silencing on apoptosis-related proteins *BCL2* and *BAX* in GBM cells

Analysis on whether *FBP1* affected the apoptosis of GBM cells showed that *FBP1* silencing changed the expressions of the pro-apoptotic protein *BAX* and the anti-apoptotic protein *BCL2* in U87 cells. Silencing *FBP1* remarkably downregulated the level of *BCL2* protein

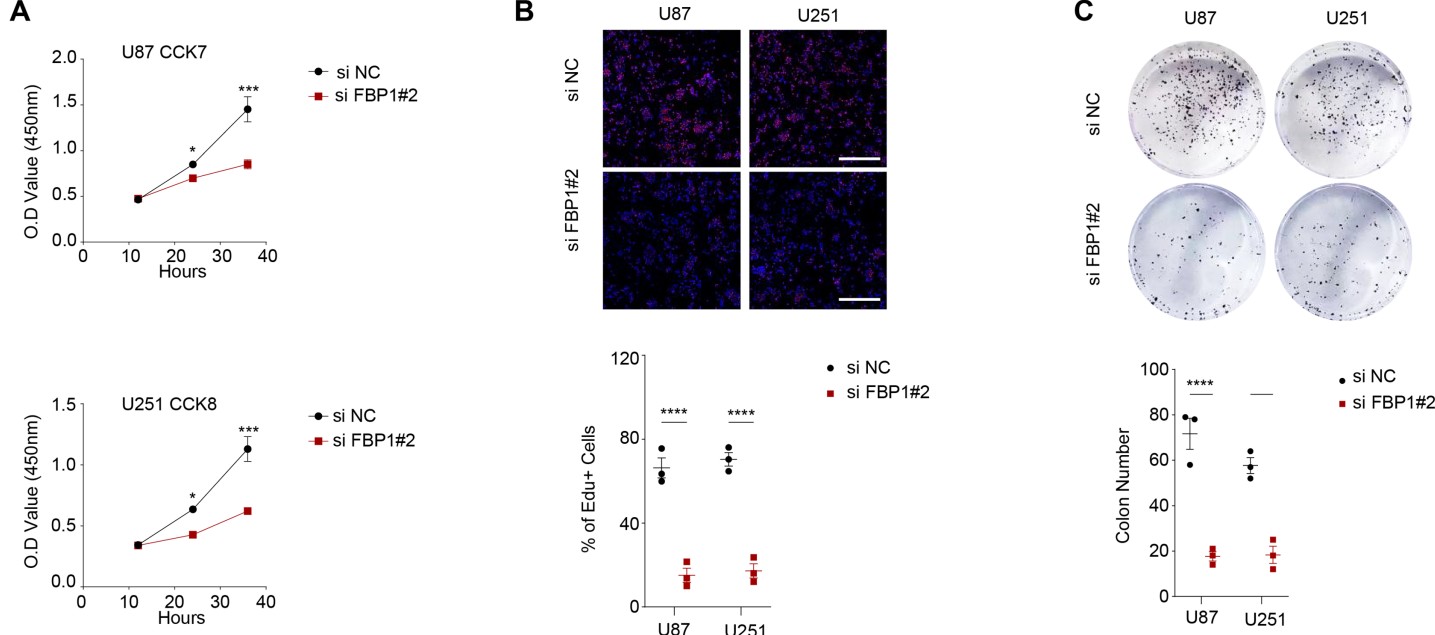

**Figure 3** **Effect of *FBP1* on the proliferative capacity of glioblastoma cells (U87 and U251).** (A–C) Based on CCK-8 (A), EdU (B) and clone formation assays (C) to assess the effect on U87 and U251 cell proliferative capacity after *FBP1* silencing. All data of three independent trials were expressed as mean ± standard deviation. The statistical significance was shown as either the asterisks (*$p < 0.05$, ***$p < 0.001$, ****$p < 0.0001$).

and significantly promoted the protein expression of *BAX* (Figs. 6E and 6F, $p < 0.05$). In U251 cells, BCL2 expression was downregulated and BAX expression was upregulated, which further validated the effect of *FBP1* silencing on the expressions of the proteins correlated with apoptosis (Figs. 6G and 6H).

## Effects of *FBP1* silencing on the M2 macrophage polarization

As shown by the results of immunofluorescence assay, silencing *FBP1* suppressed the mean fluorescence intensity of M2 macrophage indicator CD206 (Figs. 7A and 7B, $p < 0.01$), which was in accordance with a lowered degree of STAT6 phosphorylation (Figs. 7C–7F, $p < 0.001$). These results confirmed the effects of *FBP1* silencing on suppressing the M2 macrophage polarization.

## DISCUSSION

GBM is a prevalent primary intracranial malignancy with a poor prognosis and a median survival shorter than 2 years (*Tan et al., 2020*). At present, the standard therapies for newly diagnosed GBM are surgery and postoperative temozolomide in combination with radiotherapy, however, the overall prognosis of GBM patients still remains unfavorable (*Angom, Nakka & Bhattacharya, 2023*). The emergence of immunotherapy has improved the treatment outcomes of a variety of malignancies, but its effect on the malignancies within the central nervous system is weaker due to the peculiar immune microenvironment (*Ransohoff & Engelhardt, 2012*). Moreover, GBM cells will develop resistance against the standard treatments and the heterogeneity of GBM also increases the

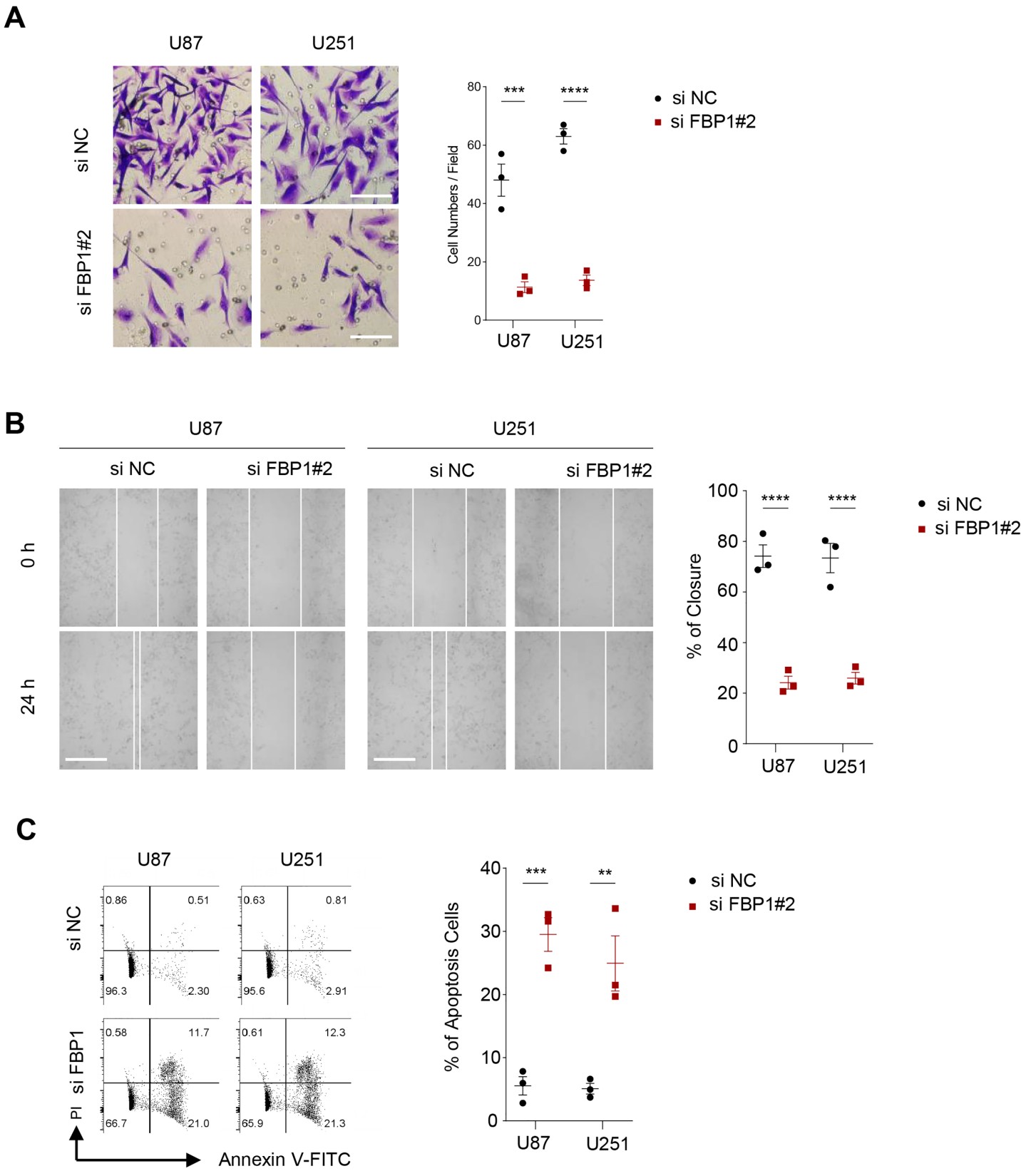

**Figure 4 Effect of *FBP1* on invasion, migration and apoptotic capacity of glioblastoma cells.** (A) The transwell-based assay to assess the effect of *FBP1* silencing on the invasive capacity of U87 and U251 cells. (B) Wound healing assay to assess the effect of *FBP1* silencing on the migratory

**Figure 4** (continued)
capacity of U87 and U251 cells. (C) Flow cytometry to assess the effect of *FBP1* silencing on the apoptotic capacity of U87 and U251 cells. All data of three independent trials were expressed as mean ± standard deviation. The statistical significance was shown as either the asterisks (**$p < 0.01$, ***$p < 0.001$, ****$p < 0.0001$).

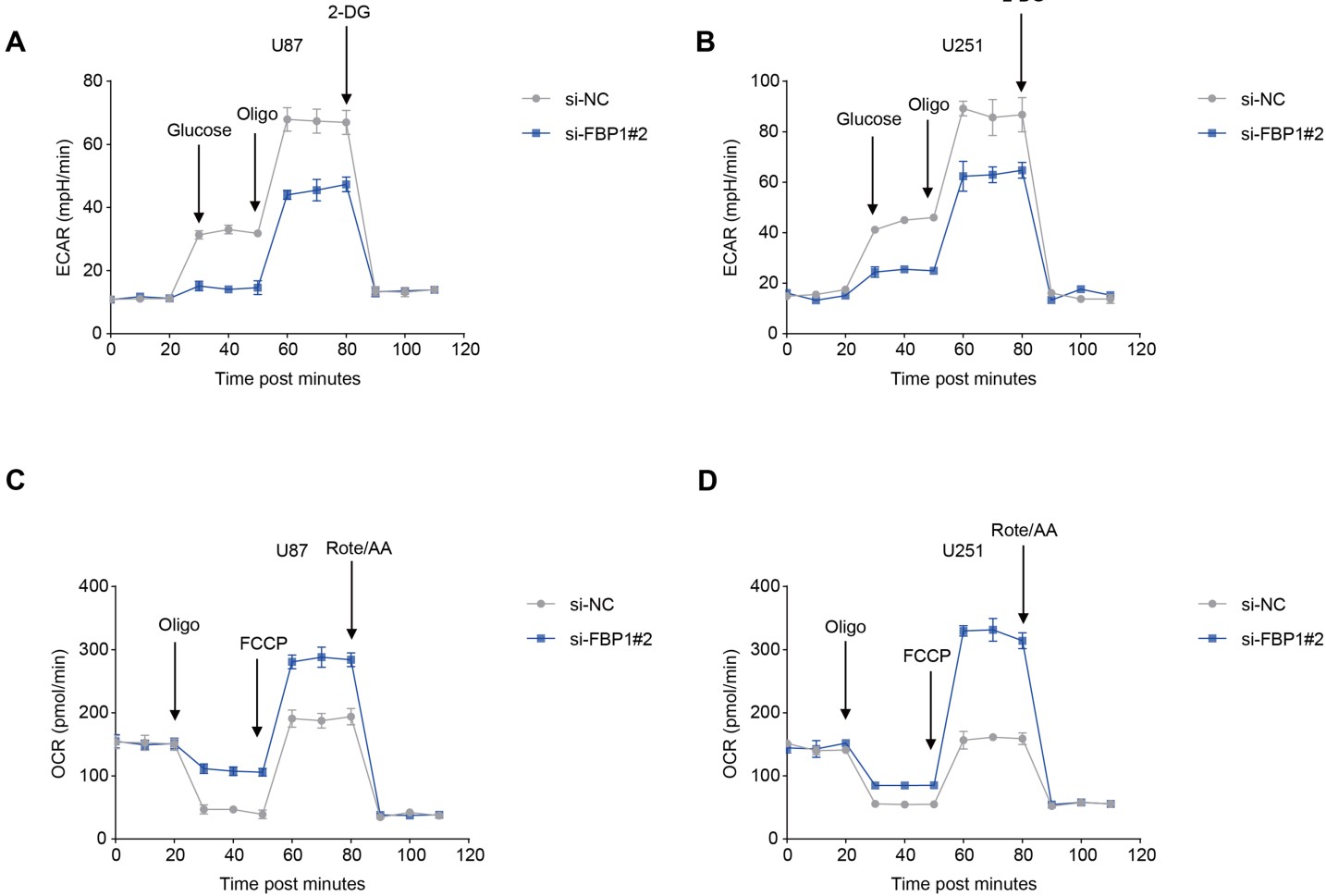

**Figure 5** Effects of *FBP1* silencing on the aerobic glycolysis of glioblastoma cells. (A and B) Quantified extracellular acidification rate (ECAR) of glioblastoma cells U87 (A) and U251 (B). (C and D) Quantified oxygen consumption rate (OCR) of glioblastoma cells U87 (C) and U251 (D). All data of three independent trials were expressed as mean ± standard deviation.

difficulties in treatment (*Loginova et al., 2024*). Therefore, though great efforts have been devoted to advance the immunotherapy and precision treatment for GBM (*Li et al., 2024*), novel effective targets are still required to improve the outcomes of GBM patients.

*FBP1* is a rate-limiting enzyme that mainly facilitates the process of gluconeogenesis and suppresses glycolysis. Loss of *FBP1* could cause lactic acidosis and hypoglycemia and even unexpected death to infants (*Cong et al., 2018*). *In vitro* genetic manipulation on *FBP1* showed that *FBP1* suppresses cell growth and triggers the production of reactive oxygen species (*Chen et al., 2011*; *Dong et al., 2013*; *Li et al., 2014*). It has been demonstrated that GBM cells are highly dependent on aerobic glycolysis, for instance, the Warburg effect, to

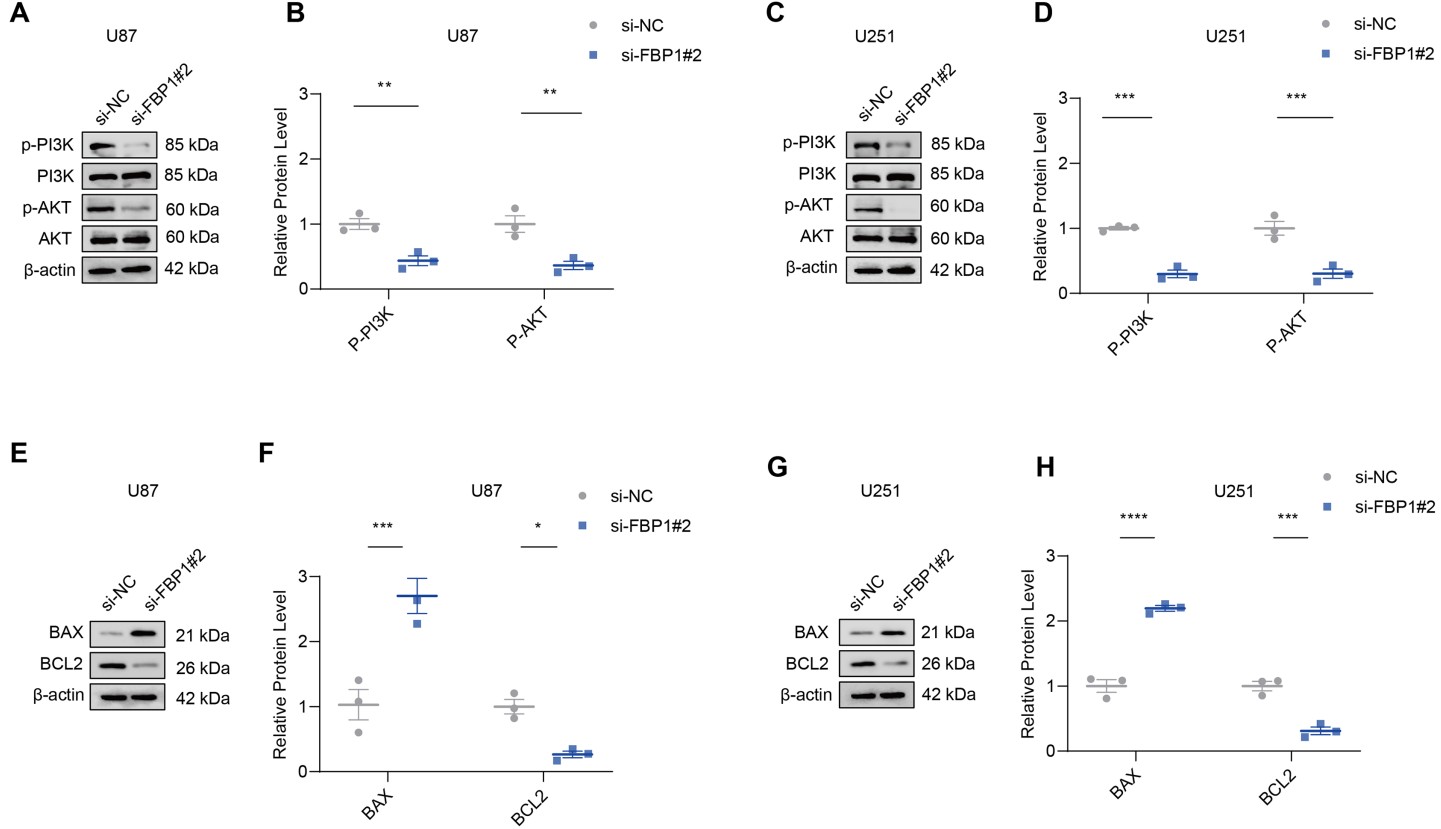

**Figure 6** Effects of *FBP1* silencing on the mediators related to PI3K/AKT pathway and BCL2 protein family. (A–D) Effects of *FBP1* silencing on the protein expressions of mediators related to PI3K/AKT pathway in glioblastoma cells U87 (A and B) and U251 (C and D). (E–H) Effects of *FBP1* silencing on the protein expressions of mediators related to BCL2 protein family in glioblastoma cells U87 (E and F) and U251 (G and H). All data of three independent trials were expressed as mean ± standard deviation. The statistical significance was shown as either the asterisks (*$p < 0.05$, **$p < 0.01$, ***$p < 0.001$, ****$p < 0.0001$).

meet their energy and metabolite requirements for rapid proliferation (*Xiao et al., 2024*). Aerobic glycolysis not only provides ATP, but also supports nucleotide and lipid synthesis through intermediate metabolites to sustain cell growth and proliferation (*Zhang et al., 2023*). Studies on the dual roles of *FBP1* in tumor progression have shown that its expression can influence tumor behavior. Specifically, manipulating *FBP1* expression can either promote or suppress tumor progression by enhancing or inhibiting glycolysis and cell growth (*Dong et al., 2013*; *Li et al., 2014*; *Hirata et al., 2016*). In glioma, *FBP1* is involved in the promoting effects of glycogen branching enzyme 1 on the aerobic glycolysis of the cancer (*Chen et al., 2023*). These results indicated that silencing *FBP1* by blocking aerobic glycolysis may suppress the malignant phenotypic characteristics of GBM cells. Previous research confirmed *FBP1* as a potential prognostic biomarker correlated with immunosuppressive tumor microenvironment (TME) in GBM (*Sun et al., 2023*). M2-type macrophages play pro-tumorigenic roles in the TME to support angiogenesis, immune escape, and tumor cell migration and invasion. Our study found that silencing *FBP1* significantly downregulated the expression of the M2-type macrophage marker CD206. It

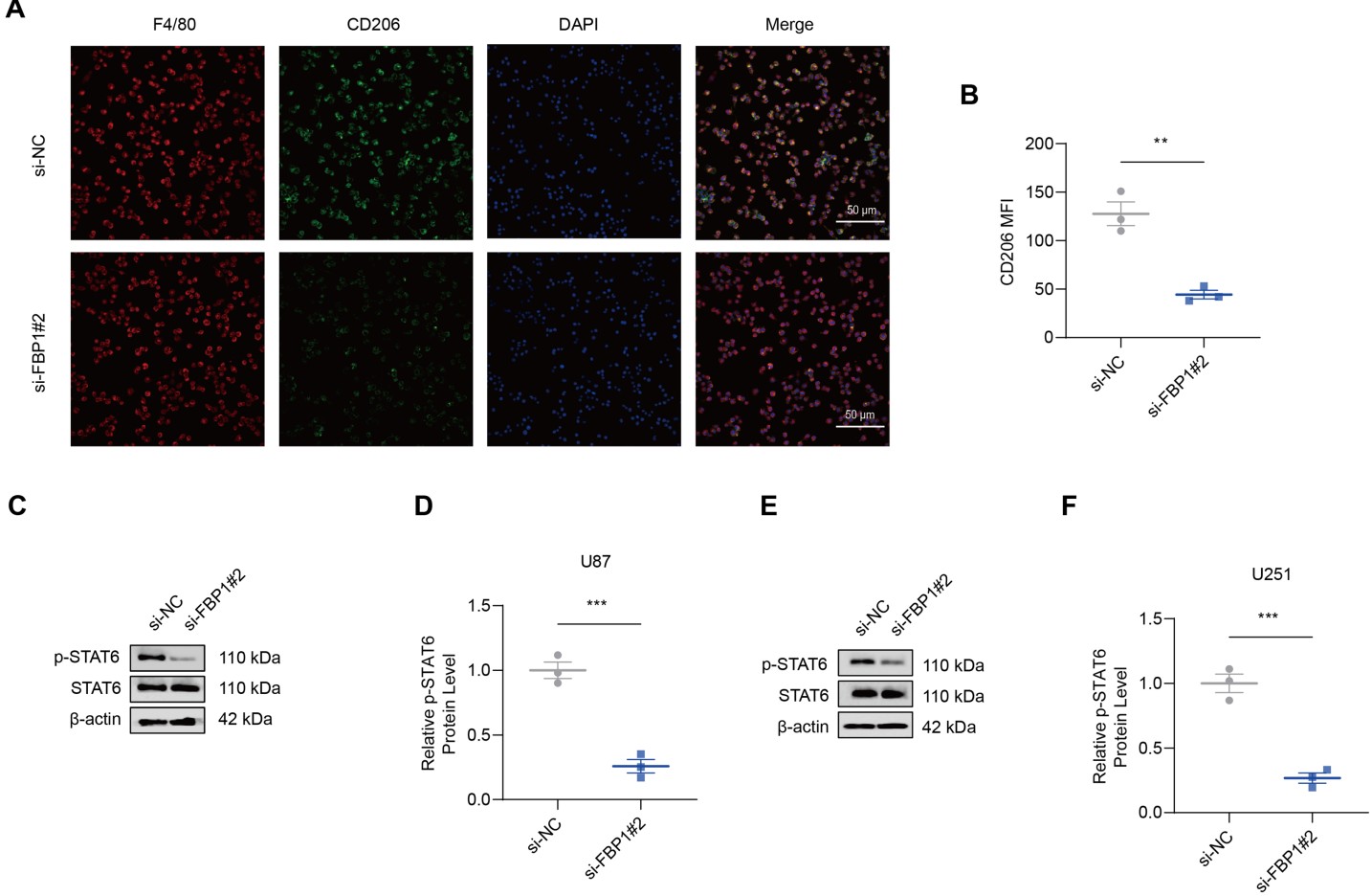

**Figure 7 Effects of *FBP1* silencing on the M2 macrophage polarization.** (A and B) Quantified mean fluorescence intensity of CD206 following the silencing of *FBP1*. (C–F) Quantified p-STAT6 protein expression in macrophage co-cultured with *FBP1*-silenced glioblastoma cells U87 (C and D) and U251 (E and F). All data of three independent trials were expressed as mean ± standard deviation. The statistical significance was shown as either the asterisks (**$p < 0.01$, ***$p < 0.001$).                         

also inhibited signaling and STAT6 phosphorylation. These results suggest that *FBP1* influences M2 polarization by regulating the STAT6 signaling pathway.

Effective and targeted chemotherapies for treating GBM require systemic analysis to explore the specific signaling pathways to characterize the driver genes in GBM development and progression (*Barzegar Behrooz et al., 2022*). The PI3K/AKT pathway plays a critical role in regulating essential biological processes, including epithelial-mesenchymal transition (EMT), cell proliferation, metabolism, and angiogenesis. Targeting specific regulatory components of this pathway may disrupt key functions of GBM cells, offering potential therapeutic opportunities (*Barzegar Behrooz et al., 2022*). Study reported that solute carrier family 25 member 32 (*SLC25A32*) promotes the malignant progression of GBM through triggering the PI3K/AKT pathway (*Xue et al., 2023*). Moreover, midkine could also activate the PI3K/AKT signaling to enhance GBM progression (*Hu et al., 2021*). NAD(P)H: quinone acceptor oxidoreductase 1 (*NQO1*) drives the aggressiveness of GBM cells *via* inducing EMT and the PI3K/AKT pathway

(*Zheng et al., 2023*). To interpret the potential effects of *FBP1* on PI3K/AKT pathway, it has been found that *FBP1* is one of the hypoxia-related genes associated with the hypoxia microenvironment in non-small cell lung cancer, which is also related to the PI3K/AKT pathway (*Zhang et al., 2021*). This study confirmed the effects of *FBP1* on PI3K/AKT pathway in GBM. According to our results, silencing *FBP1* inhibited the phosphorylation of PI3K and AKT. However, whether *FBP1* acted on the PI3K/AKT pathway to modulate the malignant phenotype and aerobic glycolysis of GBM cells required verification by further rescue assays using relevant agonists or antagonists of PI3K/AKT pathway, and this will the focus of our future study.

However, some limitations in this study should be noticed. Firstly, our study was mainly based on *in vitro* experiments, and in the future, we will conduct a comprehensive investigation of the effects of *FBP1* on the occurrence and development of GBM by establishing a mouse xenograft tumor model. Secondly, we also lacked a direct validation dataset of GBM patient samples, and the correlation between *FBP1* expression, clinicopathologic parameters, and prognosis should be analyzed using patient tissue samples. Finally, although *FBP1* affected GBM development through inhibiting the PI3K/AKT pathway and M2-type macrophage polarization, its upstream and downstream regulators as well as molecular networks were not explored in depth. To this end, further studies are encouraged to analyze the potential upstream and downstream targets of *FBP1* using transcriptomics and proteomics to reveal the regulatory network.

## CONCLUSION

To conclude, this study revealed the role of *FBP1* in GBM and the underlying mechanism based on *in vitro* cellular assays. It was found that high-expressed FBP1 was closely related to an unfavorable prognosis of GBM. Importantly, silencing FBP1 significantly suppressed GBM cell viability, proliferation, invasion, and migration capacity, inhibited aerobic glycolysis and M2-type macrophage polarization through modulating the PI3K/AKT signaling pathway. Collectively, this study revealed the potential role of *FBP1* as a key regulator of the TME and malignant phenotype of GBM, providing a potential target and theoretical basis for developing novel therapies.

## ABBREVIATIONS

| | |
|---|---|
| **GBM** | Glioblastoma |
| *BRG1* | Brahma-Related Gene-1 |
| *NRBP1* | Nuclear receptor-binding protein 1 |
| **PI3K** | Phosphoinositide 3-kinase |
| **AKT** | Protein kinase B |
| **ATP** | Adenosine triphosphate |
| **PRMTs** | Protein arginine methyltransferases |
| *POU2F2* | POU class homeobox 2 |
| *PDPK1* | Phosphoinositide-dependent Kinase-1 |
| **mTOR** | Mechanistic Target of Rapamycin |
| *FBP1* | Fructose 1,6-bisphosphatase 1 |

| EZH2 | Enhancer of zeste homolog 2 |
| --- | --- |
| H3K27me3 | Histone 3 Lys 27 trimethylation |
| PMA | Phorbol-12-myristate-13-acetate |
| CCK-8 | Cell counting kit-8 |
| OD | Optical density |
| ECAR | Extracellular acidification rate |
| OCR | Oxygen consumption rate |
| FCCP | Carbonyl cyanide-4 (trifluoromethoxy) phenylhydrazone |
| DAPI | 2-(4-Amidinophenyl)-6-indolecarbamidine dihydrochloride |
| BAX | BCL-2 associated X |
| STAT6 | Signal transducers and activators of transcription 6 |
| SLC25A32 | Solute carrier family 25 member 32 |
| NQO1 | NAD(P)H: quinone acceptor oxidoreductase 1 |

### Funding

This study was supported by a grant from the Science and Technology Project of Jinhua city of Zhejiang Province in China (Grant Nos. 2021-4-311, 2021-4-105 and 2023-4-036). The funders had no role in study design, data collection and analysis, decision to publish, or preparation of the manuscript.

### Grant Disclosures

The following grant information was disclosed by the authors:
Science and Technology Project of Jinhua city of Zhejiang Province in China: 2021-4-311, 2021-4-105 and 2023-4-036.

### Competing Interests

The authors declare that they have no competing interests.

### Author Contributions

- Weihong Lu conceived and designed the experiments, performed the experiments, analyzed the data, authored or reviewed drafts of the article, and approved the final draft.
- Guozheng Huang performed the experiments, authored or reviewed drafts of the article, and approved the final draft.
- Yihan Yu performed the experiments, analyzed the data, prepared figures and/or tables, and approved the final draft.
- Xia Zhai conceived and designed the experiments, analyzed the data, prepared figures and/or tables, authored or reviewed drafts of the article, and approved the final draft.
- Xiangfeng Zhou conceived and designed the experiments, authored or reviewed drafts of the article, and approved the final draft.

## Data Availability

The raw data is available in GitHub and Zenodo:

- https://github.com/XiangfengZhou540/Raw-data.git

- XiangfengZhou540. (2024). XiangfengZhou540/Raw-data: Updated raw data (v.1.1.1).
Zenodo. https://doi.org/10.5281/zenodo.14403400

## Supplemental Information

Supplemental information for this article can be found online at http://dx.doi.org/10.7717/peerj.18926#supplemental-information.

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
