# Peer review of "Fructose 1,6-bisphosphatase 1 is a potential biomarker affecting the malignant phenotype and aerobic glycolysis in glioblastoma"

_PeerJ, doi:10.7717/peerj.18926_

## Round 0.1 · original submission · Major Revisions

· Academic Editor

Major Revisions

Three reviewers have carefully evaluated your work and provided constructive feedback. While they acknowledge the value of your research, substantial revisions are needed to enhance the quality and clarity of your manuscript. Please carefully address all reviewers' comments, particularly focusing on the major concerns. A detailed point-by-point response to all reviewers' comments should be included with your revised manuscript.

Reviewer 1 ·

Basic reporting

Thank you for the invitation from the editor. I have carefully read the manuscript and the following are my comments:
Major comments:
1. The specific statistical results in Figure 5b, d, and Figure 5f are strikingly similar. The authors are requested to provide the original data for verification.
2. In Figure 3e, the si-FBP1#2 and si-NC show a significant discrepancy in the size of the purple staining. Is this variation due to impurities in the staining agent? If so, these results do not represent a successful experiment and should be redone and resubmitted.
4. The scope of the experiments presented in this manuscript is limited. To strengthen the study of cancer cell characteristics, it is advised to incorporate additional assays such as the Edu assay and scratch test. Furthermore, the study of apoptosis is typically integral to cancer cell research. It is suggested that the authors consider adding these experiments, time and funding permitting, to enhance the persuasiveness of their findings.
Minor comments:
1. Figure 1a lacks sufficient resolution, making it difficult to discern the details.
2. fig1 a The GBM should be circled because it is not possible to see the level of the GBM.
3. The language of the manuscript could benefit from further refinement. For instance, "Despite the great effort made" could be more succinctly phrased as "Despite significant efforts." A thorough linguistic revision of the entire manuscript is recommended.
4. The western blot image in Figure 2c is too blurry. A high-resolution image is requested.
5. The results for HEB in Figure 2a and b appear to have discrepancies and should include error bars for accuracy.
6. The images displayed in Figure 3a and b are incomplete and require amendment.
7. The image in Figure 3c is not clear enough. A high-resolution image is needed, with visible cell clusters.
8. Figure 3f's y-axis indicates "colon number," but the text describes it as an invasion assay result.
9. The meanings of si-FBP1#2 and si-NC should be clarified in the methods section.
10. The results in Figure 5a-d and e-h should have distinct subtitles, as they represent different aspects: the former pertains to pathway effects, while the latter to apoptosis effects.
11. STR authentication and mycoplasma testing results should be provided for all cell lines used.
12. What is the rationale for focusing on the PI3K/AKT pathway in this study? This pathway is quite common, and it would be beneficial to understand the specific significance of its investigation in the context of this research.
13. The figure legends should also specify the sample size, e.g., n=3.
14. Have all the results in Figure 4 undergone statistical significance testing? How are conclusions of significant differences drawn without such comparisons?

Experimental design

no comment

Validity of the findings

no comment

Reviewer 2 ·

Basic reporting

Through the bioinformatics analysis combined with various cell experiments, this present study explores the underlying mechanism of Fructose 1,6-bisphosphatase 1 (FBP1) in glioblastoma (GBM). This research demonstrates that FBP1 could act as a biomarker affecting the malignant phenotypes and aerobic glycolysis in GBM, which could provide novel insights into the clinical treatment of GBM. Overall, the experimental design of this study is logical and innovative; but before publication, the manuscript needs to be improved and revised according to the following suggestions.
1. In the introductory section, several evidences of FBP1 knockdown affecting tumorigenesis in ovarian cancer, prostate cancer, and colorectal cancer were stated. However, considering that the object of this study is GBM, it is suggested to add more introductions about the research progress of FBP1 in GBM into the manuscript.
2. At the end of the introduction, it is recommended to supplement a paragraph about the work to be carried out, the indicators to be measured and the methods to be used, as well as the significance of performing this present study.
3. The full name of the abbreviations “FBP1” (Line68) and “CCK-8” (Line100) needs to be provided in the manuscript.
4. In the Line103-104, for the colony formation assay, 400 transfected GBM cells were seeded into the 6-well plate and maintained for 14 days. What is the unit of “400”? Please clarify in the methods section.
5. What are the specific procedures and parameters of PCR operation? Please supplement in the appropriate position in the manuscript.
6. The source of the database employed for bioinformatic analysis and the specific analyses process including the software packages used require to be provided.
7. The statement of entire results section in the manuscript seems oversimplified, hence, it is recommended to describe the results in detail.

Experimental design

8. In vitro experiments of this study showed that silencing FBP1 could inhibit the aerobic glycolysis of GBM cells and the M2 macrophage polarization. What are the specific biological implications on GBM development of these results? Please discuss in detail in the discussion section.
9. The research limitations and future research directions are recommended to be presented in the last paragraph of the discussion section.
10. According to the research results of this study, how can summarize the conclusion of “the FBP1/PI3K/AKT axis was identified as a modulator of GBM progression”? Please give a reasonable explanation.

Validity of the findings

no comment

·

Basic reporting

• Language: The manuscript is written in professional English. The text is unambiguous, with appropriate terminology for an academic audience. However, some sentences in the introduction and discussion are lengthy and could benefit from concise rephrasing for enhanced readability.
• Literature context: The introduction provides relevant references to contextualize the study, citing foundational work on GBM and biomarkers like FBP1. However, recent findings could be highlighted more comprehensively to support the rationale for the study.
• Figures and tables: The figures are high-quality, well-labeled, and effectively illustrate the findings. Tables listing sequences used in the study are appropriately detailed and clear.
• Raw data availability: The authors provide access to raw data on GitHub and Zenodo, fulfilling the journal's data-sharing policy.
• Structure adherence: The manuscript adheres to PeerJ standards, with a logical structure and clear sections for methods, results, and discussion.

Experimental design

• Scope and originality: This is original primary research within the scope of the journal. The study aims to address a knowledge gap in GBM biomarker research by exploring FBP1's role in malignant phenotypes and glycolysis.
• Research question: The research question is well-defined and relevant, focusing on FBP1's potential as a prognostic marker and therapeutic target in GBM.
• Methodological rigor: The experimental design is robust, with clear descriptions of cell lines, transfection protocols, and assays (e.g., CCK-8, Transwell, and glycolysis measurements). These methods appear sufficiently detailed for replication.
• Ethical standards: There is no indication of ethical concerns related to the use of cell lines.

Validity of the findings

• Data: The data are statistically sound, with proper controls and replication. Results are presented clearly, with appropriate statistical analyses and significance reporting.
• Conclusions: The conclusions are well-supported by the data, linking FBP1 knockdown to reduced GBM cell proliferation, invasion, and aerobic glycolysis. However, the study acknowledges limitations, such as the lack of in vivo validation and clinical data, which should be addressed in future research.
• Impact: While impact is not assessed, the study contributes meaningfully to GBM research by identifying FBP1 as a modulator of GBM progression via the PI3K/AKT pathway.

Additional comments

Strengths:
• Comprehensive analysis of FBP1's role in GBM, supported by cellular assays and bioinformatics.
• Clear presentation of results with robust statistical support.
• Thorough discussion linking findings to broader implications in GBM research.
Weaknesses:
1. Lack of in vivo validation: The study relies solely on in vitro experiments, limiting the generalizability of the findings to clinical settings.
2. Mechanistic insights: The role of the PI3K/AKT pathway is suggested but not experimentally validated with rescue experiments. Future studies should include agonists/antagonists to confirm FBP1's direct involvement.

---

## Round 0.2 · Minor Revisions

· Academic Editor

Minor Revisions

Based on the reviewers' evaluations, your manuscript requires minor revisions before final acceptance. Please address the following points:

1. Please revise lengthy sentences in the introduction and discussion sections to enhance readability.
2. Consider incorporating more recent literature to strengthen the study rationale, particularly in the introduction section.

Once these minor revisions are completed, your manuscript will be ready for acceptance.

Reviewer 1 ·

Basic reporting

I have carefully read the manuscript, and although it is not 100% complete, I can see that the author has made great efforts. The current quality of the manuscript has reached a high standard, and I have no new comments.

Experimental design

no comment

Validity of the findings

no comment

Reviewer 2 ·

Basic reporting

This study explored the potential mechanism of fructose-1,6-diphosphatase-1 (FBP1) in glioblastoma (GBM) through bioinformatics analysis combined with various cell experiments. This study suggests that FBP1 can serve as a biomarker affecting the malignant phenotype and aerobic glycolysis of GBM, providing new insights for the clinical treatment of GBM. Overall, the experimental design of this study is logical and innovative; The author has adopted my suggestion and the quality of the manuscript has been significantly improved. Congratulations.

Experimental design

None

Validity of the findings

None

·

Basic reporting

Language and clarity: The manuscript is written in clear and professional English. The text is unambiguous, with appropriate terminology for an academic audience. However, some sentences in the introduction and discussion are lengthy and could benefit from concise rephrasing for enhanced readability.
Literature context: The introduction provides relevant references to contextualize the study, citing foundational work on GBM and biomarkers like FBP1. However, recent findings could be highlighted more comprehensively to support the rationale for the study.
Figures and tables: The figures are high-quality, well-labeled, and effectively illustrate the findings. Tables listing sequences used in the study are appropriately detailed and clear.The authors provide access to raw data on GitHub and Zenodo, fulfilling the journal's data-sharing policy.
Structure adherence: The manuscript adheres to PeerJ standards, with a logical structure and clear sections for methods, results, and discussion.

Experimental design

Scope and originality: This is original primary research within the scope of the journal. The study aims to address a knowledge gap in GBM biomarker research by exploring FBP1's role in malignant phenotypes and glycolysis.
Research question: The research question is well-defined and relevant, focusing on FBP1's potential as a prognostic marker and therapeutic target in GBM.
Methodological rigor: The experimental design is robust, with clear descriptions of cell lines, transfection protocols, and assays (e.g., CCK-8, Transwell, and glycolysis measurements). These methods appear sufficiently detailed for replication. There is no indication of ethical concerns related to the use of cell lines.

Validity of the findings

Data robustness: The data are statistically sound, with proper controls and replication. Results are presented clearly, with appropriate statistical analyses and significance reporting.Conclusions: The conclusions are well-supported by the data, linking FBP1 knockdown to reduced GBM cell proliferation, invasion, and aerobic glycolysis. However, the study acknowledges limitations, such as the lack of in vivo validation and clinical data, which should be addressed in future research.
Impact: While impact is not assessed, the study contributes meaningfully to GBM research by identifying FBP1 as a modulator of GBM progression via the PI3K/AKT pathway.

---

## Round 0.3 · Minor Revisions

· Academic Editor

Minor Revisions

While I noted the efforts you made to address the concerns regarding the sentence length of the Introduction and Discussion sections, there appears to be a discrepancy regarding the updated references in the Introduction section. In your response letter, you mentioned updating the references; however, these tracked updates are not visible in the revised manuscript.

Please clearly indicate which references have been updated in the Introduction section and submit the revised version. Once this minor revision is addressed, we can proceed with the final acceptance of your manuscript.

---

## Round 0.4 · accepted · Accept

· Academic Editor

Accept

I have reviewed your revision and note that you have appropriately updated reference #15 as requested, replacing the Zhang et al. (2013) citation with the more recent Li et al. (2024) publication. The change has been clearly marked in yellow in the reference section. Your manuscript is now acceptable for publication.